# Do Multilingual LLMs Think In English?

**Lisa Schut**
OATML, Department of Computer Science,
University of Oxford,
Oxford, UK.
schut@robots.ox.ac.uk

**Yarin Gal**
OATML, Department of Computer Science,
University of Oxford,
Oxford, UK.

**Sebastian Farquhar**
Google DeepMind,
London, UK.

## Abstract

Large language models (LLMs) have multilingual capabilities and can solve tasks across various languages. However, we show that current LLMs make key decisions in a representation space closest to English, regardless of their input and output languages. Exploring internal representations with a logit lens for sentences in French, German, Dutch, and Mandarin we show that the LLM first emits representations close to English for semantically-loaded words before translating them into the target language. We further show that activation steering works better for these LLMs when the steering vectors are computed in English than in the language of the inputs and outputs. This suggests that multilingual LLMs perform key reasoning steps in a representation that is heavily shaped by English in a way that is not transparent to system users.

## 1 Introduction

Large Language Models (LLMs) are predominantly trained on English data, yet are deployed across various languages, including some languages rarely seen during training. This raises an important question: *how* do LLMs operate across different languages?

LLMs are hypothesized to operate in an abstract concept space (Chris Olah, 2023; Nanda et al., 2023a; Wendler et al., 2024; Dumas et al., 2024b). From the multilingual perspective, one question is whether the concept space is language-specific or language-agnostic. We consider three different hypotheses:

1. LLMs 'operate' in a space that is English-centric (or centred on the main pretraining language)

2. LLMs 'operate' in a language-agnostic space

3. LLMs 'operate' in a language-specific space, which is determined by the input language.

We present evidence that the first hypothesis is true: LLMs reason in an English-centric way. Our work studies open-ended multi-token language generation, contrasting with prior work (Wendler et al., 2024) which found evidence for the second hypothesis in the single token context.

We analyse four open source models – Llama-3.1-70B, Gemma-2-27b, Aya-23-35B and Mixtral-8x22B– which vary in architecture and language coverage. We study three aspects of language generation:

1. **Decode the representation space** to show that LLMs make semantic decisions in English, even when prompted in a non-English language. However, non-lexical words do not appear to route through the English representation space. Fig. 1 shows the logit lens to Llama-3.1-70B as it generates the French text *Le bateau naviguait en douceur sur l'**eau au calme du lac. Le soleil ...**,* with the bold text representing Llama's output. The English translation

(a) Prompted in French, with *Le bateau naviguait en douceur sur l'*. The nouns "eau", "lac" and "soleil" are selected in English, whereas other parts of speech are not.

(b) Prompted in Dutch with *Ze telen hun eigen*. The nouns "fruit" and "vegetable", verb "kweken" and pronoun "they" are selected in English, whereas the coordinating conjunction "en" is not.

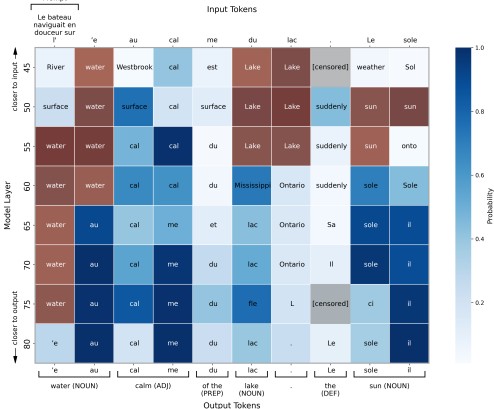

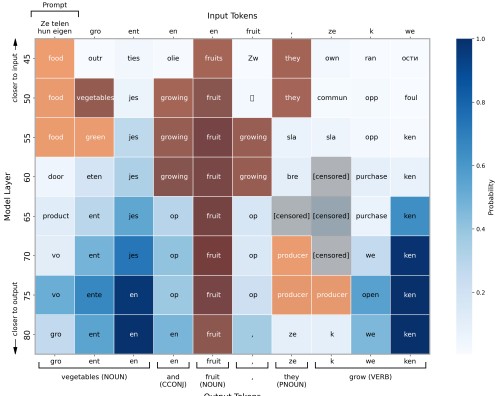

Figure 1: Logit lens applied to Llama-3.1-70B's latent space. Each row depicts the decoded latent representations for one layer and each column corresponds to the generated token. Orange boxes highlight words selected in English, darker red boxes highlight related words, while gray boxes indicate explicit terms omitted from the figure (see Appendix B.5.1).

is *The boat sailed smoothly on the **calm water in of the lake. The sun ...***. Lexical words like "water," "lake," and "sun" are selected in English, whereas grammatical elements such as "du" and "le" are not. We find that this trend holds more generally for other models (Sec. 4.1), with Aya being the least English-centric and Gemma the most English-centric.

2. **Manipulate the representations** to show that non-English sentence generations can be steered more effectively using English-derived steering vectors than those derived from the target language. This surprising result provides further evidence that LLMs rely on an English-centric conceptual space for semantic reasoning. Further, we find that the steering vectors have a relatively high cosine similarity, however, they do encode a language-specific component. We can increase the similarity of steering vectors found in different languages by nudging them towards each other using a language steering vector.

3. **Analyse the structure of the latent space.** Fact representations are shared between languages, allowing interpolation between a fact expressed in two languages while maintaining the correct answer—changing only the output language.

This English-centric behaviour of LLMs causes them to perform worse in other languages, whether in downstream tasks (Shafayat et al., 2024; Huang et al., 2023; Bang et al., 2023; Shi et al., 2022), or in fluency (Guo et al., 2024). Moreover, this impacts the fairness of these models – which currently exhibit cultural biases (Shafayat et al., 2024) – and their robustness and reliability in diverse linguistic settings (Marchisio et al., 2024; Deng et al., 2024).

## 2 BACKGROUND

### 2.1 LARGE LANGUAGE MODELS

Language models are trained to operate across different languages. Table 3 summarizes the four LLMs we study, which differ in the number of languages they were trained on. Aya-23-35B supports the widest range of languages, while Gemma-2-27b covers the fewest.

We evaluate these models across five languages, selected based on their varying levels of representation during training. English, the predominant training language, is a baseline. French and German represent high-resource, non-English languages, while Dutch and Chinese are lower-resource languages. Dutch is only a high-resource language in Aya-23-35B and therefore provides an interesting

Table 1: LLM-Insight dataset examples: sentences and prompts for the word animal.

| LANGUAGE | SENTENCE EXAMPLE | PROMPT EXAMPLE |
|---|---|---|
| ENGLISH | THE ZOO HAS A WIDE VARIETY OF ANIMAL SPECIES. | THEY ADOPTED A |
| DUTCH | DE BOERDERIJ HAD ELK TYPE HUISDIER. | IN DE DIERENTUIN ZAG IK EEN BIJZONDER |
| FRENCH | LE LION EST UN ANIMAL SAUVAGE QUI VIT DANS LA SAVANE. | IL A VU UN |
| MANDARIN | 森林中生活着许多野生动物 | 每年都会有新的 |
| GERMAN | DER ZOO BEHERBERGT VIELE FASZINIERENDE TIERE. | SIE LIEBT ES ZEIT MIT IHREM |

comparison to German due to their linguistic similarity. This analysis allows us to better understand the performance disparities across languages with varying levels of representation in training.

## 2.2 METHODS

Our goal is to understand whether LLMs have a universal representation space. To address this question, we use three mechanistic interpretability methods. The logit lens (Sec. 2.2.1) allows us to examine the internal representations, while causal tracing provides insight into where facts are encoded in the model across different languages (Sec. 2.2.2). Finally, steering vectors let us intervene on the models' internal representations (Sec. 2.2.3), which allows us to verify that the representations influence the output.

### 2.2.1 LOGIT LENS

The logit lens (nostalgebraist, 2020) decodes the internal representations of an LLM into tokens. LLMs take an input $x$ and output a probability distribution over the next token. The logit lens decodes the intermediate representation $h_l(x)$ at layer $l$ into an output token, by applying the unembedding layer:

$$\text{argmax}_t \text{softmax}(W_u h_l(\text{norm}(x))) \tag{1}$$

where $x$ is the input, $W_u$ is the unembedding matrix of the model and the subscript $t$ corresponds to the token. Fig. 1 shows the logit lens applied to Llama when generating: "Le bateau naviguiait en douceur sur l'**eau au calme du lac. Le soleil ...**". For each layer (y-axis) and token position in the generation (x-axis), a token is decoded from the internal representation. The decoded tokens from the middle layers onward are more interpretable, whereas early layers are less interpretable.

### 2.2.2 CAUSAL TRACING

Causal tracing (Meng et al., 2022; Vig et al., 2020) uses causal mediation analysis to identify where facts are stored within a network. The method compares corrupted hidden states – where the information necessary to retrieve the fact has been removed – with clean hidden states –that successfully output the fact. The difference in the output probabilities of the target token in the two forward passes is the average indirect effect (AIE). This approach allows us to identify the part of the network that encodes the fact. Further details can be found in B.1.

### 2.2.3 STEERING VECTORS

Steering vectors (Subramani et al., 2022; Turner et al., 2023; Panickssery et al., 2024) are used to nudge the behavior of the LLM in the desired direction. The main idea is to add activation vectors during the forward pass of a model to modify its behavior: $h_l(x) \leftarrow h_l(x) + \gamma v_l$, where $v_l$ is the steering vector, and $\gamma \in \mathbb{R}^+$ is a scalar hyperparameter. Steering vectors are used to nudge the output of the LLM in the desired direction. For example, if we want the output to contain more "love", we can compute a steering vector as $v_l = h_l(\text{love}) - h_l(\text{hate})$. Further details can be found in Subramani et al. (2022) and Turner et al. (2023).

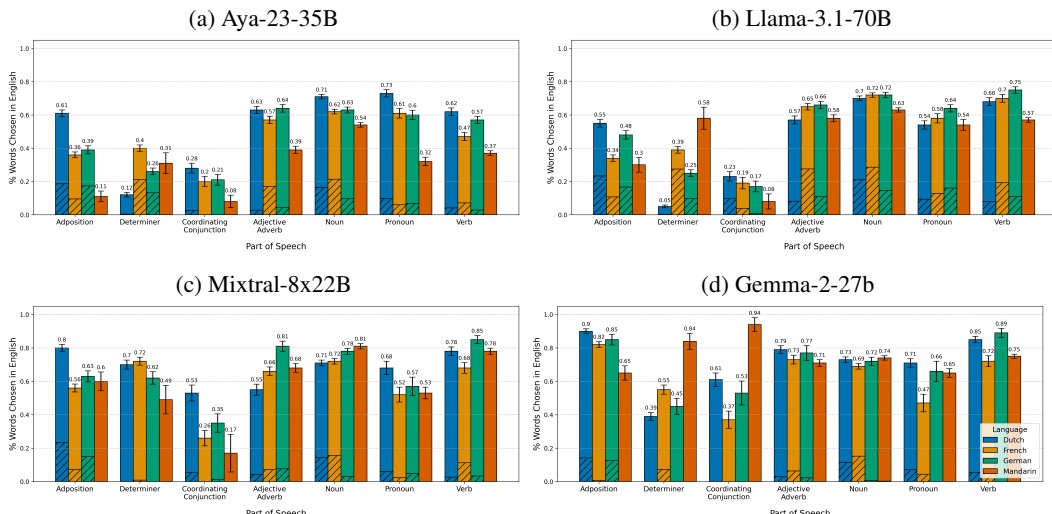

Figure 2: Logit lens analysis of LLMs routing through English. Each plot shows the proportion of words routed through the English representation space for each model. The shaded bars indicate the portion explained by homographs – words that are spelled the same in English and the specified language. Overall, the degree of English-routing depends on the model: less diverse pretraining leads to more English-routing. Similarly, most routing occurs for lexical words.

## 3 DATASETS

**LLM-Insight** We created a dataset to analyze the behavior of LLMs, which we will release alongside this paper. The dataset is specifically designed to study steering in LLMs.

It includes 72 target words, each paired with 10 prompts and 10 sentences in English, Dutch, French, German and Mandarin. Table 1 shows a sample of the data. The prompts are designed so the word could appear as the next token, but the prompts are also sufficiently open-ended so that semantically unrelated words can be used to complete the sentence. For example "They adapted a" can be completed with the word "animal" as well as "daughter".

The sentences can be used to find steering vectors. Some words in the dataset naturally form pairs that can be used to create steering vectors, such as the words "good" and "bad". For words without a natural pairing, such as "thermodynamics", we provide a general set of sentences as the counter set to create the steering vector. Further details on the dataset can be found in B.3.

**City facts (Ghandeharioun et al., 2024)** We use this dataset to investigate how facts in different languages are encoded in LLMs. The task is to provide the capital city of a given country. For example, when prompted with "The capital of Canada is", the model should output "Ottawa". This allows us to identify where in the network Ottawa is encoded. To analyse cross-lingual representations, we augment the dataset by translating these facts into German, Dutch, and French.

## 4 EXPERIMENTS

We want to understand whether LLMs process prompts differently depending on the output language. First, we analyze the latent space to find that LLMs make semantic decisions that are more closely aligned with the English representation space (Sec. 4.1). Next, we show that we can steer activations better when using English steering vectors (Sec. 4.2). Lastly, in Sec. 4.3, we show that the representations of facts are shared across languages, but have an English-centric bias when decoded.

### 4.1 INSPECTING THE LATENT SPACE OF LLMs USING THE LOGIT LENS

**Qualitative Examples** To build an intuition on how LLMs operate when prompted in different languages, we analyze their latent space using the logit lens, which decodes the internal representations.

Table 2: English-routing in LLMs: percentage of generated words that are routed through English. Aya-23-35B shows the least routing behavior, whereas Gemma-2-27b shows the most routing behavior.

| MODEL | DUTCH | FRENCH | GERMAN | MANDARIN | AVERAGE |
|---|---|---|---|---|---|
| GEMMA-2-27B | $0.72 \pm 0.01$ | $0.67 \pm 0.01$ | $0.72 \pm 0.01$ | $0.71 \pm 0.01$ | $0.70 \pm 0.00$ |
| MIXTRAL-8X22B | $0.69 \pm 0.01$ | $0.63 \pm 0.01$ | $0.71 \pm 0.01$ | $0.69 \pm 0.01$ | $0.68 \pm 0.01$ |
| LLAMA-3.1-70B | $0.51 \pm 0.01$ | $0.57 \pm 0.01$ | $0.58 \pm 0.01$ | $0.55 \pm 0.01$ | $0.55 \pm 0.00$ |
| AYA-23-35B | $0.58 \pm 0.01$ | $0.49 \pm 0.01$ | $0.50 \pm 0.01$ | $0.41 \pm 0.01$ | $0.50 \pm 0.00$ |

In Fig. 1, nouns and pronouns are routed through English, whereas the coordinating conjunction is not. Similarly, Fig. 1 shows the logit lens applied to Llama-3.1-70B for the Dutch prompt *Ze telen hun eigen*. The noun "fruit", verb "kweken" and pronoun "they" are all routed through the English words, whereas the coordinating conjunction "en" is not routed through the English word "and". Interestingly, the word growing appears in the latent space several tokens before "kweken" is generated, suggesting that the LLM may plan words in advance in English, which builds on Pal et al. (2023)'s finding that LLMs encode future tokens in the latent space.

**Quantitative Evaluation**    The qualitative examples shown in Fig. 1 suggest that the part of speech determines whether LLMs employ English routing. To investigate this, we prompt each LLM to generate 720 sentences. For each generated word, we evaluate whether the English equivalent of a word appears in the latent space. For example, in Fig. 1 (right), for the word groenten, we check whether the English equivalent, vegetables, appears in the decoded latent space. We then aggregate the results across different parts of speech. Further implementation details are provided in B.5.

Fig. 2 shows the results for Aya-23-35B, Llama-3.1-70B, Mixtral-8x22B and Gemma-2-27b. Each bar shows the percentage of words that route through the English representation space. The shaded part shows the proportion explained by cross-lingual homographs, words that are the same in English and the specified language (e.g., water in English and Dutch). For homographs, it is not possible to disambiguate whether the word routes through English.

In general, lexical words – nouns and verbs – are often chosen in English. These parts of speech influence the semantic meaning of the sentence. Other parts of speech, such as adpositions, determiners and compositional conjugates are infrequently routed through English in Aya-23-35B and Llama-3.1-70B.

The degree of English routing is model-dependent, as shown in Table 2. One explanation is the degree of multilingualism in the pre-training data – with more multilingual models, such as Aya-23-35B, routing less through English, in contrast to the least multilingual model, Gemma-2-27b, which routes the most through English. However, this does not account for the differences observed between Mixtral-8x22B and Llama-3.1-70B, for which French and German are both high-resource languages. Another possible explanation is model size. Smaller models, such as Mixtral-8x22B and Gemma-2-27b, route through English more frequently than larger models, potentially due to their more limited representation space

## 4.2   CROSS-LINGUAL STEERING

Our experiments in Sec. 4.1 suggest that LLMs may first select topic words in an English representation space, before translating them into the output language in the later layers. To further investigate this hypothesis, we evaluate whether non-English model outputs can be modified using English steering vectors.

More concretely, we test whether we can steer models to generate a sentence in a specified output language using two types of steering vectors: (1) **topic steering vector** – encourages the LLM to generate a sentence with the given topic, such as animals; (2)**language steering vector** – encourages the model to generate text in the desired output language. We evaluate the effectiveness of steering across various topics and prompts, using the LLM-Insight dataset (see Sec. 3). We evaluate steering as successful if the generated sentence includes the target word associated with the steering vector while avoiding output collapse – incoherent sentences or stuttering.

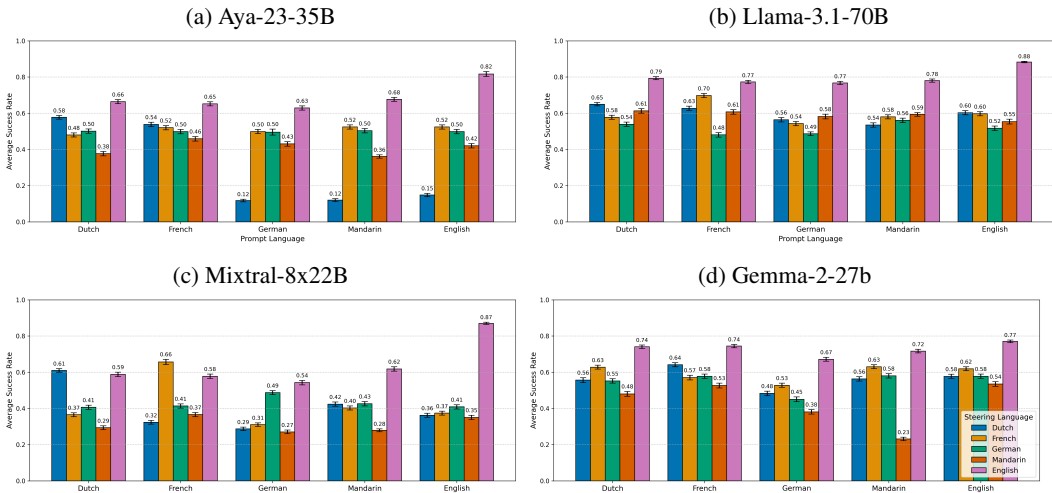

Figure 3: Cross-Lingual Steering LLMs: The language on the x-axis is the prompt and the desired output language, while the color of each bar indicates the language used to generate the topic steering vectors.

Fig. 3 shows results when steering different LLMs. In general, we observe that English steering vectors perform the best – outperforming steering vectors generated using the desired output language. This suggests that the representation space is not universal – if it were, we would expect the cross-lingual performance to be roughly equal across languages. Instead, this supports the hypothesis that these models select these words in English.

**How similar are the steering vectors generated in different languages?**   The steering vectors for the same concepts generated in different languages have a relatively high cosine-similarity, particularly in the early middle layers (see B.8). However, the steering vectors are not language-agnostic – part of the dissimilarity of the steering vectors can be attributed to the difference in the language used to generate the vectors. This further supports the argument that the representation space is not universal.

### 4.3   INVESTIGATING THE REPRESENTATION SPACE

In this section, we study *how* cross-lingual facts are encoded relative to each other using the city facts dataset (see Sec. 3). First, we perform causal tracing to determine whether facts in different languages are encoded in the same part of the model. Fig. 4 shows the causal traces for Aya-23-35B (see B.9 for other LLMs). We find that facts are generally localized in similar layers, regardless of the language.

Next, we want to understand if the representation of a fact is shared across different languages. In particular, if we have the same fact in two different languages, such as English and Dutch, can we decompose the representation as follows:

$$h(\text{capital of Canada}) = h_{Ottawa} + h_{English} \qquad (2)$$
$$h(\text{hoofdstad van Canada}) = h_{Ottawa} + h_{Dutch}, \qquad (3)$$

where $h$ represents a vector in the latent space. If the above equations hold, we may be able to interpolate between the facts:

$$\alpha h(\text{hoofdstad van Canada}) + (1 - \alpha)h(\text{capital of Canada})$$
$$= h_{Ottawa} + \alpha h_{Dutch} + (1 - \alpha)h_{English}$$

If we pushforward the interpolated hidden state, and the output is correct, then this suggests that we may be able to disentangle the language and semantic context.

We find that we can interpolate between the hidden states without significant changes in accuracy; the accuracy generally interpolates between the accuracies of the two languages (see B.10). Furthermore, we find that models have a propensity to answer in English, where propensity is measured as

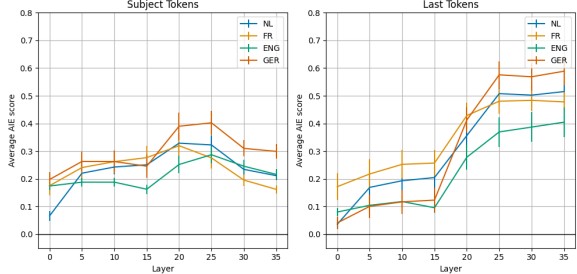

Figure 4: Causal traces for the City Facts dataset in Aya-23-35B. The AIE scores are similar across different languages, suggesting that facts are localized in the same area of the model.

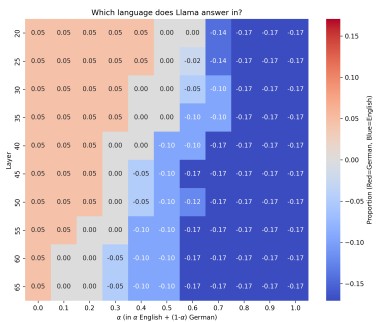

Figure 5: Relative propensity of Llama-3.1-70B to answer in German (red) vs English (blue).

the difference in the probability of the English token versus the token in the other language (see Figure **??**, and B.10). This provides further evidence that the models likely operate in an English-centric space.

## 5 RELATED WORK

We can think about understanding a model from two perspectives:

- an **internal perspective**, focused on analyzing the model through the latent space and operations performed inside of the model. Examples of questions include: how do models represent knowledge across different languages? How does a model retrieve facts?

- an **external perspective**, focused on analyzing the model output. For example, how well do models perform in different languages?

Having a unifying theory that combines both perspectives is important – the internal perspective helps us understand the mechanisms underlying behavior, while the external perspective examines the real-world impact of that behavior. Below, we summarize the research on multilingual language models from both perspectives.

### 5.1 HOW DO LLMs OPERATE INTERNALLY?

The current main theory in mechanistic interpretability suggests that there are three general phases in the forward pass of an LLM Chris Olah (2023); Nanda et al. (2023a); Wendler et al. (2024); Dumas et al. (2024a); Fierro et al. (2025):

1. **Detokenization**: In this phase, individual tokens are combined into abstract units that the model uses for analysis. These units can be referents – for example, Nanda et al. (2023a) found evidence that the tokens [Michael] and [Jordan] are combined into a unit representing the basketball player Michael Jordan. Similarly, these units can encode instructions, as shown by Dumas et al. (2024a), where the model extracts the target language during translation tasks in these layers.

2. **Processing**: In this phase, the model processes or reasons over abstract units. For instance, this stage may involve tasks like fact recall Geva et al. (2023); Nanda et al. (2023a).

3. **Selecting the output**: In this phase, the model selects the output. This may involve selecting the correct attribute Nanda et al. (2023a), mapping an abstract concept to the corresponding word in the target language Wendler et al. (2024) and/or selecting the correct token for the intended word.

In the context of multilingual models, an important question is whether the concept space (in phase 2) is universal. Here, *universal* means the representation is *shared* across languages, i.e., the representation for "cat" (cat in English) and "kat" (cat in Dutch) is the same.

One stream of research argues that the concept space is *universal*. When analyzing the latent space with the logit lens, Wendler et al. (2024) find that the concept space is language-agnostic, but more closely aligned with the English output space. In their follow-up work, Dumas et al. (2024a) used tracing to find further evidence of language-agnostic representations of concepts.[1]. Concurrent to this work, Brinkmann et al. (2025) showed that models share morpho-syntactic concept representations across languages.

Other researchers found that the concept space is biased towards the training-dominant language. Using the logit lens, Zhong et al. (2024) focused on Japanese, showing that Swallow (Fujii et al., 2024), which is fine-tuned on Japanese, and LLM-jap Aizawa et al. (2024), pre-trained on Japanese and English, are Japanese-centric. Fierro et al. (2025) found that subject enrichment – retrieving attributes about the subject – is language-agnostic whereas object extraction is language-dependent, in EuroLLM (Martins et al., 2024), XLGM (Lin et al., 2022) and mT5 (Xue et al., 2021).

Overall, our findings suggest the presence of a language-specific latent space. Specifically, our results indicate that the concept space is largely English-centric, consistent with prior work Zhong et al. (2024); Wu et al. (2024), where English is the dominant training language. However, unlike previous studies, we uncover additional nuance: non-lexical words are not necessarily routed through the English representation space. Moreover, we find that the behavior varies across models. One potential explanation is that we focus on open language generation – which is different from prior work that predominantly focuses on single-token and translation tasks. We provide a more in-depth discussion in Section 7.

## 5.2 Multilingual LLM behavior

Here, we summarize how the internal mechanisms of LLMs affect performance.

**Performance**  The performance of multilingual language models varies across languages Shafayat et al. (2024); Huang et al. (2023); Bang et al. (2023); Shi et al. (2022); Ahuja et al. (2023), often performing best in English. This can even be leveraged to improve performance in other languages, for example, through cross-lingual chain-of-thought reasoning Chai et al. (2024), or by modifying prompts, such as using multilingual instructions or asking the LLM to translate the task into English before completing it (Zhu et al., 2023; Etxaniz et al., 2023).

**Fluency and language confusion**  Marchisio et al. (2024) show that English-centric models are prone to language confusion, i.e., providing the answer in the incorrect language. Moreover, even when LLMs output text in the correct language, they can produce unnatural sentences in other languages, akin to an accent Guo et al. (2024).

**Bias and culture**  LLMs tend to be biased toward certain cultures, performing better when working with facts originating from Western contexts Naous et al. (2024); Shafayat et al. (2024), and falling short when answering questions on other cultures Chiu et al. (2024). Liu et al. (2024) investigate the cultural diversity of LLMs using proverbs and find that these models often struggle to reason with or effectively use proverbs in conversation. LLM's understanding appears limited to memorization rather than true comprehension, creating a culture gap when translating or reasoning with culturally specific content across languages.

## 6 Conclusion

Our results provide evidence that semantic decisions in LLMs are predominantly made in a representation space close to English, while non-lexical words are processed in the prompt language. However, we find that this behavior varies across models, likely due to differences in multilingual proficiency and model size. The English-centric behavior is further validated by our findings that steering non-English prompts using vectors derived from English sentences is more effective than those from the prompt language.

Exploring the structure of the latent space, we find that factual knowledge across languages is stored in roughly the same regions of the model. Interpolating between the latent representations of these

---

[1]They used tracing to mitigate potential shortcomings of cosine-similarity (Steck et al., 2024)

facts in different languages preserves predictive accuracy, with the only change being the output language. This suggests that facts encoded in different languages likely share a common representation. However, when interpolating, we find that model output is most frequently in English, further underlining the English-centric bias of the latent space.

The English-centricity of the latent space is consistent with prior observations about LLM behavior. In particular, Etxaniz et al. (2023) found that instructing LLMs to first translate a non-English prompt into English improves model performance. However, this bias can be detrimental. If the latent space is English-centric, this may lead the LLMs toward exhibiting Western-centric biases (Naous et al., 2024; Shafayat et al., 2024).

## 7 DISCUSSION

There are currently two perspectives in interpretability research on concept representations in multilingual models: (1) concept representations are universal; and (2) concepts have language-centric representations, where the language is the training-dominant language. Our work aligns more closely with the second perspective, as well as a third perspective – namely, that LLMs encode language-specific representations, where the language is the input/output language. Below, we discuss how the different theories may be reconciled.

Wendler et al. (2024) and Dumas et al. (2024a) argue that the concept space is universal, but likely more aligned with the English output space. However, our findings contest this conclusion, as we find that interventions in the latent space are more effective when using English text, even when the target language is not English. If the concept space were truly universal, we would expect interventions using all languages to perform equally well. Our findings are consistent with concurrent work by Wu et al. (2024), who similarly find that steering using English performs comparably to, or slightly better than, the target language.

One possible way to reconcile the two theories is via the difference between concepts that are encoded and concepts that are used (as discussed in Brinkmann et al., 2025). There may be multiple representations of any given concept (Hase et al., 2023; McGrath et al., 2023), or a concept may be represented in an LLM but not used during generation. Wendler et al. (2024) focus more on the encoding of concepts, whereas our work focuses more on the generation of text.

An alternative explanation is that different behavior is captured in the tasks. In Wendler et al. (2024); Dumas et al. (2024a), the tasks are designed to generate a single token. In this setting, the task is to select the correct token, and we expect a high probability mass on a single token. In contrast, we focus on a more open-ended setting where there are several different possible continuations. These two settings are inherently different, leading to different conclusions about the behavior of LLMs. Even within the same task of fact retrieval, prior work found that different components of the forward pass are language-specific and language-agnostic (Fierro et al., 2025).

More generally, the open-generation setting allows us to analyze different parts of speech. This leads to the second main difference in conclusions, which is that LLMs encode language-specific representations. For semantically loaded words, we find evidence that the latent space is English-centric (in LLMs where English is the dominant training language). This is consistent with one line of prior work, which generally focuses on nouns (Wu et al., 2024; Zhong et al., 2024). However, we find that the same pattern does not hold for non-lexical words.

This is in contrast to concurrent work by Brinkmann et al. (2025), who showed that models share morpho-syntactic concept representations across languages in Llama-3-7b and Aya-8B. In line with their previous work (Wendler et al., 2024), they argue that the representations are universal. While our high-level conclusions differ, our findings also support the hypothesis that smaller models emit more shared representations than larger models, which permit more language-specific representations.

In summary, our findings indicate that the extent to which representations are shared across languages is more nuanced than previously thought. Contrasting our work with previous work suggests that the task and model size likely influence the observed behavior. Fully understanding these nuances is important to ensure the fairness and robustness of LLMs.

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

## IMPACT STATEMENT

Large Language Models (LLMs) are increasingly deployed across a wide range of applications, making it crucial to understand and evaluate their performance to ensure both safety and fairness. A key characteristic of LLMs is their English-centric nature, which influences their behavior, as shown in this paper. This behavior impacts performance, and addressing these impacts is essential to promote equitable and reliable outcomes in diverse linguistic and cultural contexts.

## A   APPENDIX

## B   LIMITATIONS

Our work provides evidence suggesting that MLLMs primarily operate in English. Below, we outline potential limitations and directions for future research.

**Tokenization**   Sentences in different languages often vary in tokenization length (Rust et al., 2021; Muller et al., 2021; Petrov et al., 2024), which complicates cross-lingual comparisons. We provide heuristics (e.g., for causal tracing, which operates on a per-token level) to allow us to compare the results when tokenization lengths vary. However, future work should further investigate tokenization remains an important consideration for the development of future interpretability methods designed to be used across multiple languages.

**Language confidence and confusion**   Models often assign higher probabilities to outputs in certain languages, which can affect analyses, such as causal tracing by requiring higher noise levels. Similarly, models often exhibit language confusion (Marchisio et al., 2024), continuing to respond in English even when prompted in other languages. Both factors influence our analysis. We can mitigate some issues associated with the first problem – e.g., in causal tracing, we ensure the probabilities all fall below a specified threshold when a prompt is noised. However, we do not actively address language confusion, as doing so could alter the natural behavior of the LLMs, which we aim to understand.

**Factors affecting interpretability methods**   Interpretability methods are influenced by various factors. Steering performance, for example, depends on the intrinsic steerability of a prompt (Turner et al., 2023; Tan et al., 2024). To address this, we designed a custom dataset that, to the best of our knowledge, is equally steerable across all languages. Another challenge is that steering could push activations outside the expected data distribution, leading to unintended outputs. To mitigate this, we checked for stuttering in the generated outputs. However, further work is needed to deepen our understanding of steering mechanisms and to develop more robust evaluation procedures.

**Other Methods**   Exploring alternative methods could provide valuable insights. For example, sparse autoencoders (SAEs) (Olshausen & Field, 1997; Hinton & Salakhutdinov, 2006; Templeton et al., 2024) are a popular interpretability tool. However, training SAEs for each layer is computationally expensive and beyond our computational budget. While some pre-trained SAEs are available, they are predominantly trained on English data, which introduces biases we aim to avoid (Lieberum et al., 2024).

## B.1   Causal Tracing

Causal tracing (Meng et al., 2022; Vig et al., 2020) use causal mediation analysis to identify where facts are stored within a network. For example, imagine that we want to find where the fact "The capital of Canada is Ottawa" is represented in an LLM. We could prompt the model with "The capital of Canada is" to find where "Ottawa" is stored in the network. There are two main steps in causal tracing:

1. **corrupt the signal**: destroy the information so that model no longer outputs the fact.
2. **restore the signal**: determine where in the network the representation need to be restored so that the LLM can recover the correct output.

Let $e^{\text{clean}} \in \mathbb{R}^{m,d}$ be the embedding of the prompt "The capital of Canada is", where $m$ is the number of tokens and $d$ is the embedding dimension. In the first step, the information is "destroyed" by adding noise to embedding of the subjects token:

$$e_j^{\text{corrupted}} = \begin{cases} e_j^{\text{clean}} + \varepsilon & \text{if token } j \text{ is a subject token} \\ e_j^{\text{clean}} & \text{otherwise} \end{cases} \tag{4}$$

where $\varepsilon$ is noise sampled from an isotropic Gaussian distribution, and $e^{\text{corrupted}}$ is the corrupted embedding. We pushforward corrupted embeddings $e^{\text{corrupted}}$ through the network to obtain the probability that the model outputs Ottawa, $p[\text{Ottawa}|e^{\text{corrupted}}]$.

Next, we want to find out which part of the hidden states encodes the relevant information to restore the correct output. At a given layer $l$ in the network, we 'restore' part of the corrupt hidden state by copying back part of the clean hidden state $h$ at position $p$:

$$h_{j,l}^{\text{restored}} = \begin{cases} h_{j,l}^{\text{clean}} & \text{if } j = p \\ h_{j,l}^{\text{corrupted}} & \text{otherwise,} \end{cases} \tag{5}$$

where the hidden state $h_l^{\text{clean}} = [h_{1,l}^{\text{clean}}, \dots, h_{m,l}^{\text{clean}}]$ is obtained by pushing the original embeddings, $e^{\text{clean}}$, through the network.

Finally, we propagate $h_l^{\text{restored}}$ through the remaining layers produce the output probability $p[\text{Ottawa}|h_{l,p}^{\text{restored}}]$. The difference $p[\text{Ottawa}|h_{l,p}^{\text{restored}}] - p[\text{Ottawa}|h_l^{\text{corrupted}}]$, measures the importance of layer $l$ and token position $p$ in encoding a fact. Through this approach, causal tracing helps identify which parts of the representation are sufficient to retrieve the correct output.

## B.2   LLM Training Data Languages

Table 3 summarizes the languages the different models are trained on.

## B.3   LLM-Insight Dataset

Our goal is to generate a dataset that can be used for cross-lingual interpretability. We wanted a dataset that can be used to introspect LLM internal representations and analyse LLM behaviour

Table 3: LLMs: High resource training languages

| Model | Languages |
|---|---|
| LLAMA-3.1-70B (DUBEY ET AL., 2024) | ENGLISH, GERMAN, FRENCH, ITALIAN, PORTUGUESE, HINDI, SPANISH, AND THAI |
| MIXTRAL-8X22B-V0.1 (JIANG ET AL., 2024) | ENGLISH, FRENCH, ITALIAN, GERMAN, AND SPANISH |
| AYA-23-35B (ARYABUMI ET AL., 2024) | ARABIC, CHINESE (SIMPLIFIED AND TRADITIONAL), CZECH, DUTCH, ENGLISH, FRENCH, GERMAN, GREEK, HEBREW, HINDI, INDONESIAN, ITALIAN, JAPANESE, KOREAN, PERSIAN, POLISH, PORTUGUESE, ROMANIAN, RUSSIAN, SPANISH, TURKISH, UKRAINIAN, AND VIETNAMESE |
| GEMMA-2-27B (TEAM, 2024) | ENGLISH |

when the internal representations representations are intervened on. Additionally, the dataset focuses on open-ended sentence generation rather than being restricted to specific tasks like fact recall or sentiment analysis, as text generation is an important real-world application.

### B.3.1 TEXT GENERATION

We use GPT-4o to generate sentences and prompts. For each target word, we generate:

- 10 unique sentences containing a version of the word – for example, for the verb '(to) see', a suitable sentence is 'She saw a bird in the sky.'
- a list containing the version of the word used in each sentence. In the previous example, the version of the word is 'saw'.
- 10 unique prompts, designed to be completed with the target word.
- a list containing a version of the word used in each sentences

We instruct GPT-4o to generate prompts that cane be completed with the target word, as well as semantically distinct words. However, we observe that the model sometimes produces sentences and prompts that do not meet the criteria.

An example of a sentence that does not meet the criteria is:

> **Target word**: bouquet (boeket in Dutch)
> **Sentence**: Het boeket was gevuld met levendige rozen en lelies.
> **Translation**: The bouquet was filled with live roses and lilies.

The issue with this sentence is its unnatural phrasing—the word "live" is not typically used in this context.

An example of a prompt that does not meet the criteria is:

> **Target word**: money
> **Prompt**: He went to the bank to withdraw

In this case, the only plausible continuation is "money." While the prompt is coherent, it lacks the open-endedness needed to analyze how interventions influence model behavior.

An example of a well-constructed prompt is:

> **Target word**: bus
> **Prompt**: She took a

This can be completed with the intended word "bus", as well as semantically different alternatives such as "walk" or "long road trip".

To ensure data quality, we asked native speakers to review and correct the data. The original version of the data and the corrections are provided in the dataset.

B.3.2 DATASET SUMMARY

We selected words that vary in the number of tokens (in non-English languages), whether the word is a homograph with the English version of the word, and the part of speech. Table 4 summarizes the words used.

Table 4: Word Translations Across Languages

| Word | English | Dutch | French | German | Mandarin |
|---|---|---|---|---|---|
| animal | animal, animals | dier, dieren | animal, animaux | Haustier, Tier, Tiere | 动物 |
| bad | bad | slecht, slechte | mal, mauvais, mauvaise, mauvaises | schlecht, schlechte, schlechten | 不好, 坏了 |
| ballet | ballet | ballet | ballet | Ballett | 芭蕾 |
| bank | bank | bank | banque | Bank | 银行 |
| beautiful | beautiful | mooi, mooie | beau, bel, belle, magnifique | schön, schöne, schönen | 美丽 |
| big | big, tall | groot, groots, grote | grand, grande, grands, gros | große, großen, großer, großes | 大 |
| bouquet | bouquet | boeket | bouquet | Strauß | 花束 |
| brother | brother | broer | frère | Bruder | 哥哥, 弟弟, 兄弟 |
| bus | bus | bus | bus | Bus | 公交车 / 巴士 |
| cat | cat | kat | chat | Katze, Katzen, Mutterkatze | 猫, 小猫, 流浪猫 |
| centre | centre | centrum | centre | Zentrum | 中心, 市中心, 研究中心, 社区中心, 艺术中心 |
| chair | chair | stoel | chaise | Stuhl, Stuhls, Stühlen | 椅子, 椅子, 摇椅 |
| chauffeur | chauffeur, driver | chauffeur | chauffeur, chauffeuse | Chauffeur | 司机 |
| child | child | kind | enfant | Kind | 孩子 |
| club | club | club, vereniging | club | Club | 俱乐部 |
| cold | cold | ijskoud, kou, koud, koude | froid, froide, froides | kalt | 冷, 寒冷 |
| computer | PC, computer, laptop, machine, rig, system | computer | ordinateur | Computer, Computern, Laptop | 电脑 |
| culture | culture, cultures | culturen, cultuur | culture, cultures, la culture | Kultur | 文化 |
| day | day | dag | jour, journée | Tag | 天, 日子 |
| dog | dog | hond | chien | Hund | 小狗, 狗 |

| Word | English | Dutch | French | German | Mandarin |
|------|---------|-------|--------|--------|----------|
| drink | drink | drank, drankje, drinken | boire, boisson, drinken | trinken | 饮料 |
| eat | eat | eten | dîner, manger, repas | essen | 吃 |
| fast | fast | fast, snel, snelle | rapide, vite | schnell | 快 |
| film | film | film, films | film | Film, film | 电影, 影片, 纪录片 |
| food | cuisine, cuisines, dish, food, meals | gerecht, gerechten, voedsel | cuisine, nourriture | Essen, Futter, Nahrung | 食物 |
| fruit | fruit | fruit | fruit, fruits | Frucht, Früchte, Obst | 水果 |
| garage | garage | garage, parkeergarage | garage | Garage | 车库 |
| give | give | geven, helpen | dire, donnent, donner, partager | geben | 给 |
| goal | goal | doel, doelpunt | but, objectif | Tor, Ziel, Ziele | 球门, 目标 |
| gobbledygook | buzzwords, gibberish, jargon, nonsense | gebazel, geheimtaal, jargon, koeterwaals, onzin, retoriek, waanzin, wartaal | baragouin, bla-bla, charabia, galimatias, gargouilloux | Kauderwelsch | 废话, 难懂的术语 |
| good | delicious, excellent, fun, good, great, helpful | goed, goede | bien, bon, bonne, bons | gut | 好 |
| hand | hand | hand, handen | main | Hand, Hände | 手 |
| happy | happy | blij | content, contente, contents, enthousiaste, gais, heureux, joyeuse | froh, glücklich | 快乐, 高兴, 高兴, 快乐 |
| horse | horse, pony | paard, paarden | cheval | Pferd, Pferde | 马 |
| hot | hot | heet, hete | chaud, chaude, chaudes | heiß | 热 |
| incomprehen-sibility | incomprehen-sibility | onbegrijpe-lijkheid | incompréhen-sibilité | Unverständlichkeit | 不可理解, 难以理解 |
| information | information | informatie | information, informations | Information, Informatio-nen | 信息 |
| land | land | land, landen | atterrir, campagne, nature, terrain, terrains, terre, territoire | Land | 土地, 土地, 国家 |

| Word | English | Dutch | French | German | Mandarin |
|---|---|---|---|---|---|
| machine | device, equipment, machine, maker, system | machine | machine | Maschine | 机器 |
| menu | menu | menu, menukaart | menu | Menü | 菜单 |
| money | money | geld | argent | Geld | 钱 |
| no | no | nee | non | nein | 不 |
| please | please | alsjeblieft | s'il te plaît, s'il vous plaît | bitte | 请 |
| police | police | politie | police | Polizei | 警察 |
| radio | radio | radio, radiozender | fréquence, radio | Radio | 广播, 收音机 |
| read | read | lezen | lire | lesen | 阅读 |
| room | room | kamer | chambre | Zimmer | 房间 |
| sea | sea | zee | mer, zoo | Meer | 大海, 海, 海洋, 海浪, 海风 |
| see | see | zien | voir | sehen | 欣赏, 看, 观察 |
| serendipity | serendipity | toeval, toevalstreffer | chance, coïncidence, hasard, sérendipité, éventualité | glückliche Zufälle, glücklicher Zufall | 机缘巧合 |
| sister | sister | zus, zusje | soeur | Schwester | 姐妹 |
| sleep | asleep, sleep | slaap, slapen | coucher, dormir, s'assoupir, se coucher, se reposer, sommeil, somnoler | Schlaf, schlafen | 睡眠, 睡觉 |
| slow | slow, slowly | langzaam, langzame | lent, lente, lentement, lentes, lents | langsam | 慢 |
| small | compact, little, small, tiny | klein, kleine | petit, petite | klein | 小 |
| speak | speak | spreken | communiquer, parler, s'exprimer | sprechen | 发言, 表达, 讲话, 说, 说话 |
| supermarket | grocer, grocery, market, store, supermarket | supermarkt | supermarché | Supermarkt | 超市 |
| table | table | tafel | table | Tisch | 台, 桌, 桌子 |
| take | take | maken, neemt, nemen, zorgen | prendre | mitnehmen, nehmen | 取, 带, 拿 |
| taxi | taxi | taxi | taxi | Taxi | 出租车 |
| thermodynamics | thermodynamics | thermodynamica | thermodynamique | Thermodynamik | 热力学 |

Table 5: Word overlap between languages

|  | ENGLISH | MANDARIN | GERMAN | DUTCH | FRENCH |
|---|---|---|---|---|---|
| ENGLISH | 75 | 0 | 0 | 14 | 16 |
| MANDARIN | 0 | 75 | 0 | 0 | 0 |
| GERMAN | 0 | 0 | 75 | 0 | 0 |
| DUTCH | 14 | 0 | 0 | 75 | 8 |
| FRENCH | 15 | 0 | 0 | 8 | 75 |

Table 6: Llama-3.1-70B tokenization statistics

| LANGUAGE | 1 | 2 | 3 | 4 | 5+ | MEAN |
|---|---|---|---|---|---|---|
| ENGLISH | 55 | 15 | 2 | 2 | 1 | 1.39 |
| MANDARIN | 35 | 18 | 5 | 3 | 3 | 1.80 |
| GERMAN | 17 | 30 | 13 | 2 | 2 | 2.12 |
| DUTCH | 19 | 29 | 11 | 2 | 3 | 2.09 |
| FRENCH | 20 | 32 | 7 | 3 | 2 | 2.03 |

| Word | English | Dutch | French | German | Mandarin |
|---|---|---|---|---|---|
| tour | tour | excursie, rondleiding, tour, tournee | Tour, tour, visite | Tour | 旅行, 游览 |
| water | water | water | eau | Wasser | 水 |
| write | write | schrijven | écrire | schreiben | 写, 写字 |
| yes | yes | ja | oui | ja | 对, 是 |

We selected words that varied in the number of tokens used to represent the words, and selected some which were the same as English words. Table 5 summarizes the word overlap across the different languages. Tables 6, 7, 8 and 9 summarize the average tokenization lengths of the words in different languages and models. The average number of tokens per word in English is lower than other languages.

Table 7: Gemma-2-27b tokenization statistics

| LANGUAGE | 1 | 2 | 3 | 4 | 5+ | WEIGHTED MEAN |
|---|---|---|---|---|---|---|
| ENGLISH | 66 | 6 | 1 | 1 | 1 | 1.20 |
| MANDARIN | 50 | 7 | 3 | 3 | 1 | 1.41 |
| GERMAN | 33 | 24 | 5 | 2 | 0 | 1.62 |
| DUTCH | 34 | 23 | 4 | 2 | 1 | 1.64 |
| FRENCH | 39 | 20 | 2 | 2 | 1 | 1.53 |

Table 8: Mixtral-8x22B tokenization statistics

| LANGUAGE | 1 | 2 | 3 | 4 | 5+ | MEAN |
|---|---|---|---|---|---|---|
| ENGLISH | 65 | 3 | 3 | 2 | 2 | 1.31 |
| MANDARIN | 0 | 22 | 22 | 4 | 16 | 3.52 |
| GERMAN | 15 | 30 | 13 | 2 | 4 | 2.25 |
| DUTCH | 17 | 29 | 12 | 3 | 3 | 2.19 |
| FRENCH | 21 | 28 | 8 | 5 | 2 | 2.11 |

Table 9: Aya-23-35B tokenization statistics

| LANGUAGE | 1 | 2 | 3 | 4 | 5+ | MEAN |
|---|---|---|---|---|---|---|
| ENGLISH | 59 | 11 | 3 | 2 | 0 | 1.31 |
| MANDARIN | 47 | 11 | 1 | 3 | 2 | 1.47 |
| GERMAN | 20 | 35 | 7 | 1 | 1 | 1.88 |
| DUTCH | 29 | 24 | 7 | 1 | 3 | 1.83 |
| FRENCH | 32 | 25 | 4 | 1 | 2 | 1.70 |

## B.4 PARTS OF SPEECH ANALYSIS

In this experiment, we analyse how often a word is first 'selected' in English, for each part of speech. To identify the part of speech, we used spacy models (Honnibal & Montani, 2017). To identify English words, we use enchant.Dict("en_US"). We use nl_core_news_sm, de_core_news_sm, fr_core_news_sm and zh_core_web_sm. In general, we can use spaces to identify words in sentences. For Mandarin, we use the package jieba.

## B.5 LOGIT LENS QUANTITATIVE EVALUATION

To evaluate whether a word is chosen in English, we use GPT-4o. We considered alternative evaluation procedures. We tested various translation packages but found issues with both word- and sentence-level approaches. When used on a word-level, this caused problems with colexification and did not allow for close synonyms often only providing a single translations per word. When using translation on a sentence level, it was difficult to map tokens to each word (due to changes in the sentence structure). We also considered WordNet (Miller, 1994), but it only covers nouns, verbs, adjectives, and adverbs, making it unsuitable for other parts of speech. Ultimately, we chose GPT-4o and manually verified 100 samples to ensure the evaluation was accurate.

Table 10: Part of Speech Abbreviations, Terms, and Examples

| ABBREVIATION | TERM | EXAMPLES |
|---|---|---|
| ADJ | ADJECTIVE | |
| ADP | ADPOSITION | IN, TO, DURING |
| ADV | ADVERB | VERY, EVERYWHERE |
| AUX | AUXILIARY | HAS (DONE), WAS (DONE) |
| CCONJ | COORDINATING CONJUNCTION | AND, OR, BUT |
| DET | DETERMINER | THEIR, HER, SOME |
| INTJ | INTERJECTION | OUCH |
| NOUN | NOUN | PLACE, THING, IDEA |
| NUM | NUMBER | 10, 200 |
| PRON | PRONOUN | HE, SHE, THEY |
| PROPN | PROPER NOUN | SPECIFIC NAME, PLACE |
| SCONJ | SUBORDINATING CONJUNCTION | THAT, IF, WHILE |
| SYM | SYMBOL | |
| VERB | VERB | SEE, RUN |

We ask GPT-4o to score words as follows:

- 5: An exact translation.
- 4: A close synonym.
- 3: A word with a similar but distinct meaning.
- 2: A word whose meaning is at best weakly related.
- 1: A word whose meaning is not related.

When a word receives a score of 4 or higher, we evaluate the word as chosen in English.

An example of the command we use is:

Below, you will be given a reference word in Dutch and a context (i.e., phrase or sentence) in which the word is used. You will then be given another list of English words or subparts of words/phrases.
You should respond with the word from the list that is most similar to the reference word, along with a grade for the degree of similarity.
Special Note on Contextual Translations: If an English word could form a common phrase or idiomatic expression that accurately translates the reference word, it should be rated highly. For example, if a phrase like "turned out" perfectly matches a Dutch verb, the word "turned" alone would receive a high score due to its idiomatic fit.
Special Note on Tenses: Do not penalize for different tenses. For example, the word 'want' matches 'wilde' and should receive a 5.

Degrees of Similarity: Similarity should be evaluated from 1 to 5, as follows:
5: An exact translation.
4: A close synonym.
3: A word with a similar but distinct meaning.
2: A word whose meaning is at best weakly related.
1: A word whose meaning is not related.

Consider the following examples:

**Example 1**
Reference Word in Dutch: 'waarop' Context: 'Ze had een hekel aan de manier waarop hij zijn'
English Word List: ['hicks', 'mild', 'rut', 'sens', 'spiral', 'hometown', 'how', 'manner', 'van', '101', 'ward']

Analysis: 'waarop' means "on which" in Dutch. The word 'how' is most similar to this in the list, while the other options are unrelated.
Answer Word: 'how'
Similarity Score: 4 - a close synonym

**Example 2**
Reference Word in Dutch: 'bleek'
Context: 'Ze adopteerde een zwerfdier, maar het bleek een wolf te zijn'
English Word List: ['cup', 'freed', 'freeman', 'laurent', 'turns', 'turned', 'van', '348', 'i', 'ken', 'oms']

Analysis: 'bleek' means "turned out" in Dutch, making 'turned' the most similar option.
Answer Word: 'turned'
Similarity Score: 5 - an exact translation

**Example 3**
Reference Word in Dutch: 'vaas'

Table 11: Frequency of explicit words decoded in the latent space across LLMs. Llama-3.1-70B has the highest proportion of explicit terms.

| MODEL | EXPLICIT WORDS (%) | | | | |
|---|---|---|---|---|---|
| | ENG | FR | NL | DE | ZH |
| LLAMA-3.1-70B | 6.25 | 11.25 | 18.35 | 8.13 | 7.19 |
| MIXTRAL-8X22B | 1.56 | 3.91 | 5.21 | 4.53 | 10.19 |
| AYA-23-35B | 2.50 | 2.97 | 3.44 | 2.89 | 2.69 |
| GEMMA-2-27B | 0.00 | 0.31 | 0.31 | 0.39 | 0.38 |

Context: 'Ze schikte een prachtig boeket bloemen in een vaas.'
English Word List: ['tucker', 'van', 'container', 'opp', 'van', 'vessel', '-g', '-t', '397', 'art', 'as', 'ed', 'ion', 'let']

Analysis: 'vaas' means "vase" in Dutch. The word 'vessel' is somewhat similar, as vases are vessels for holding items like flowers.
Answer Word: 'vessel'
Similarity Score: 3 - a word with a similar but distinct meaning

**Example 4**
Reference Word in Dutch: 'werd'
Context: 'Ze ging geld opnemen bij de bank en werd overvallen.'
English Word List: ['dee', 'lafayette', 'bank', 'bu', 'herself', 'kw', 'met', 'ramp', 'return', 'returning', '113', '347']

Analysis: 'werd' means 'was' in Dutch. None of these words are related.
Answer Word: None
Similarity Score: 1 - a word whose meaning is not related

**Example 5**
Reference Word in Dutch: 'vrienden'
Context: 'Ze bracht het weekend door met haar vrienden in een huisje in de Ardennen.'
English Word List: ['sag', 'sat', 'tween', 'bro', 'families', 'family', 'her', 'herself', 'mo', 'own', 'parents', 'weekend', '666', 'elf']

Analysis: 'vrienden' means "friends" in Dutch. The closest word here is 'families', which is weakly related but distinct.
Answer Word: 'families'
Similarity Score: 2 - a word whose meaning is at best weakly related

The examples are complete. Now it is your turn. The reference word will be in Dutch, and you must find the most similar English word and assess the degree of meaning similarity on a scale from 1 to 5.

The commands for other languages are similar, but adapted to provide examples in the language.

### B.5.1 EXPLICIT TEXT

### B.6 OTHER LANGUAGE-SPECIFIC PHENOMENA

We observe explicit vocabulary in the latent space of LLMs (examples can be found in Appendix B.5.1). Table 11 shows the frequency of vulgar words in the latent space, with Llama-3.1-70B showing the highest count. This model is safety-tuned in eight languages (Dubey et al., 2024), including English, German, and French, but not Dutch. This may suggest that explicit terminology is a language- and model-specific feature.

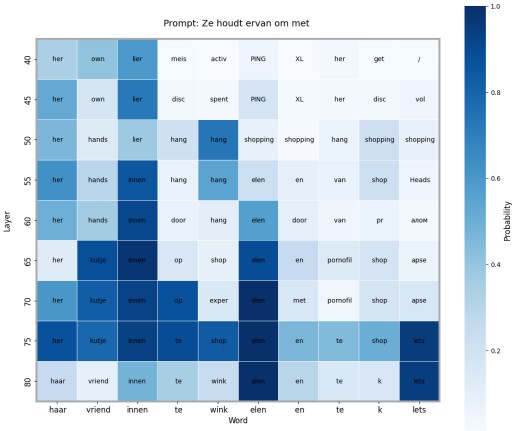

Figure 6: Example 1: Logit Lens applied to Llama-3.1-70B.

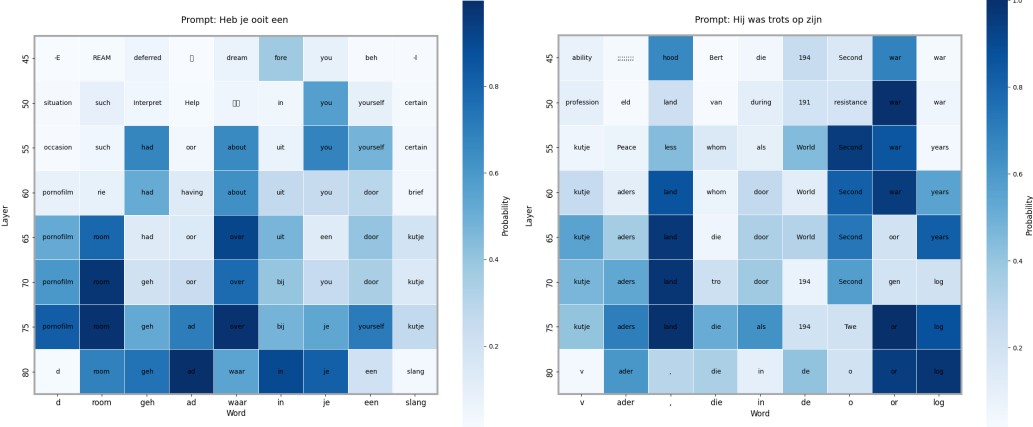

Figure 7: Logit Lens applied to Llama-3.1-70B.

Terms such as kutje (pussy), pornofilm (porn film), lul (dick), and knull (fuck) appear in various contexts, including inappropriate sentences about children. For example, in Figure 6 during the generation:

> Ze houdt ervan om met **haar vriendinnen te winkelen en te klets(en)** ...
> English translation: She enjoys shopping and talking with her friends ...

We find the explicit words 'kutje' and 'pornofil(m)' when decoding the latent space using the logit lens. This behavior is consistent across other examples (see Figure 7).

### B.7 STEERING

For the steering experiment, we use the LLM-Insight dataset. We compute two steering vectors:

- a topic steering vector: this is a steering vector that captures the intended topic. For example, for the topic 'love, we create a steering vector that is $v_l^t = h_l(\text{love}) - h_l(\text{hate})$, where $h_l$ is the hidden state in layer $l$.
- a language steering vector: we add the a steering vector that captures the intended output language. For example, for the target language Dutch, we can create a steering vector $v_l^l = h_l(\text{Dutch}) - h_l(\text{English})$.
- For each steering vector, we take the difference between sets of sentences containing the topic.

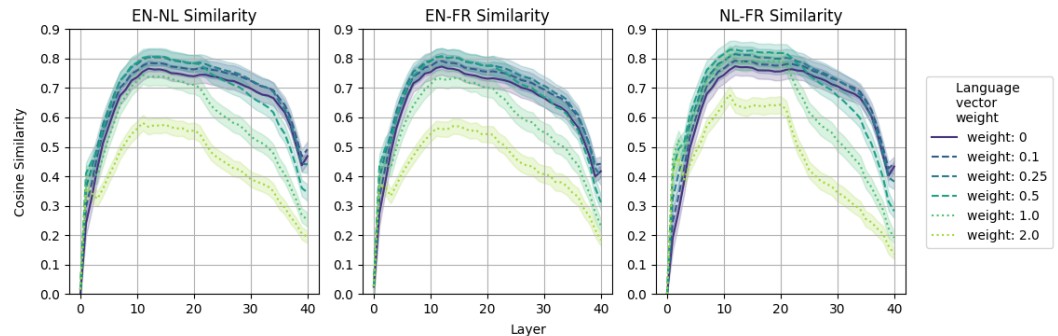

Figure 8: Cosine distance between steering vectors in Aya-23-35B.

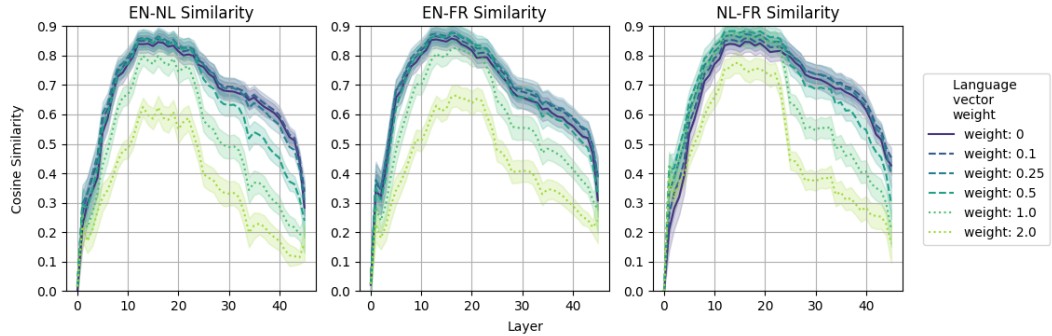

Figure 9: Cosine distance between steering vectors in Gemma-2-27b.

Currently, we consider steering successful if (1) the generated sentence contains the target word and (2) does not lead to output collapse (stuttering). We set the steering vector weights by using a hold-out set of 5 words (50 prompts). We found that 5 was optimal for the topic steering vector, and 10 was optimal for the language steering vector. For the layers, we considered every 5-th layer of the model for the topic steering vector. We considered every 2nd layer for the language vector. We reported the results across the best layers. On average, we found that 20-40 % of layers allowed for successful steering, with English steering vectors being the least sensitive to layer selection.

### B.8 COSINE SIMILARITY OF STEERING VECTORS

To analyze the geometry of the latent space, we compute both topic vectors and language vectors for our dataset. We track the cosine similarity of these steering vectors across different layers and

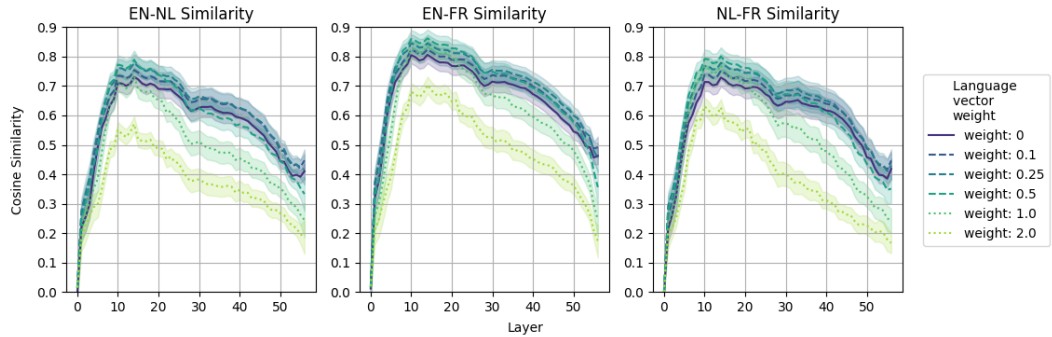

Figure 10: Cosine distance between steering vectors in Mixtral-8x22B.

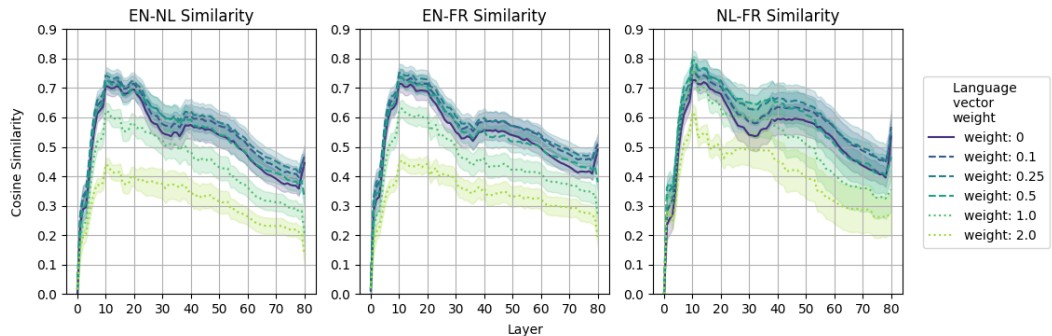

Figure 11: Cosine distance between steering vectors in Llama-3.1-70B.

models, providing insights into how topics and languages are represented internally. Specifically, we plot the cosine similarities of topic vectors derived from different languages. In Figures 8, 9, 10 and 11 are the plots for Aya-23-35B, Gemma-2-27b, Mixtral-8x22B, and Llama-3.1-70B, respectively.

We find that topic vectors maintain a high cosine similarity of approximately 0.8 across languages. The similarity can be increased by incorporating the corresponding language vector, suggesting an interaction between topic and language-specific representations.

### B.9   CAUSAL TRACING

Figures 12 and 13 show the causal traces, averaged over different country-city pairs for Aya-23-35B, Llama-3.1-70B, Mixtral-8x22B and Gemma-2-27b. Across all models, we find that facts are generally localized in similar layers, regardless of the language. Two main traces emerge: a mid-layer trace on the subject token(s), which may correspond to entity resolution and a later trace when attribute recollection occurs (as suggested by Nanda et al. (2023b)). Overall, these plots suggest that facts are approximately stored in the same parts of the model.

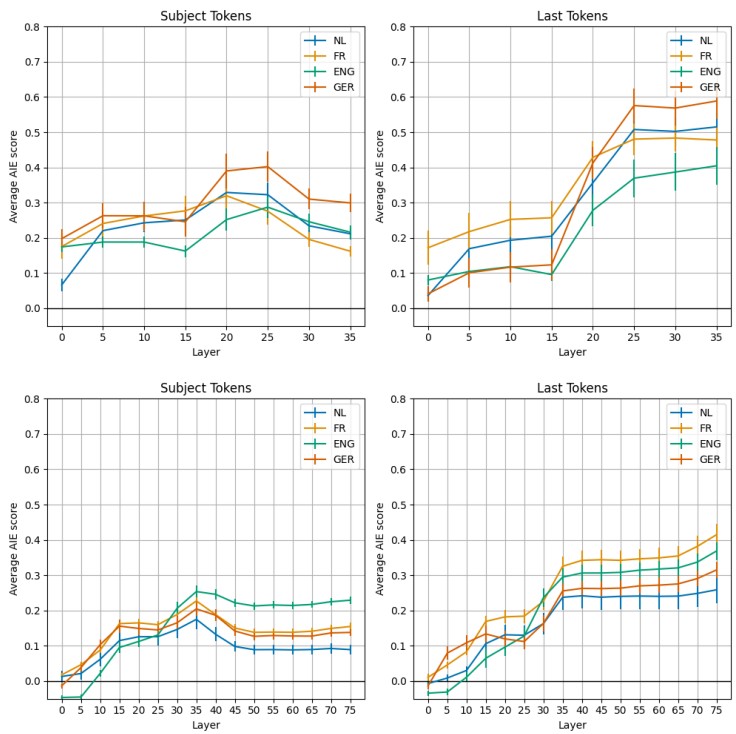

Figure 12: The causal traces of the city facts in Aya-23-35B (top) and Llama-3.1-70B(bottom).

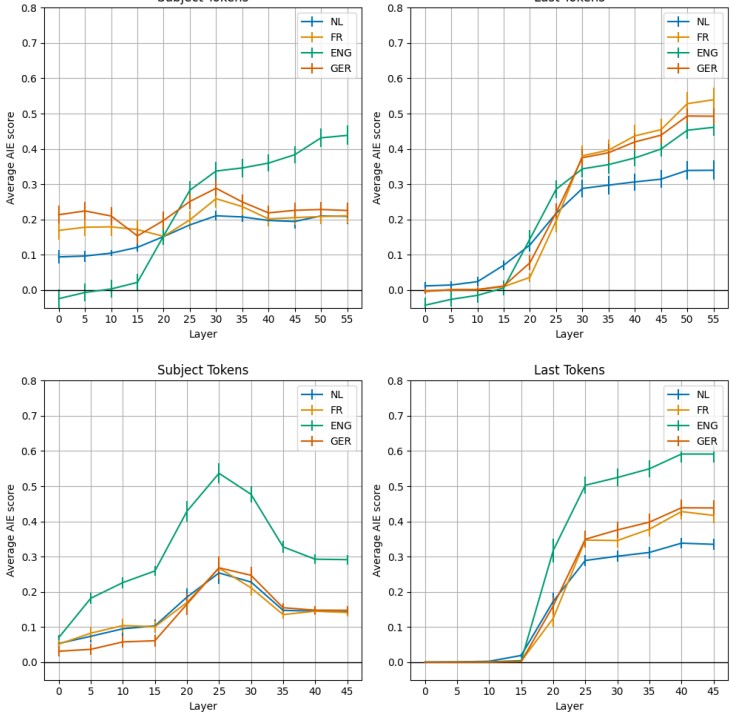

Figure 13: The causal traces of the city facts in Mixtral-8x22B (top) and Gemma-2-27b (bottom).

## B.10 HIDDEN STATE INTERPOLATION (WITH INSTRUCTIONS)

We include instructions as otherwise the LLMs often describe the city, rather than provide the city. E.g., "The capital of Canada is beautiful". For most models, the accuracy when interpolating between the hidden states is between the performances in the two languages. Interestingly, we observe a propensity of models to answer in a specific language.

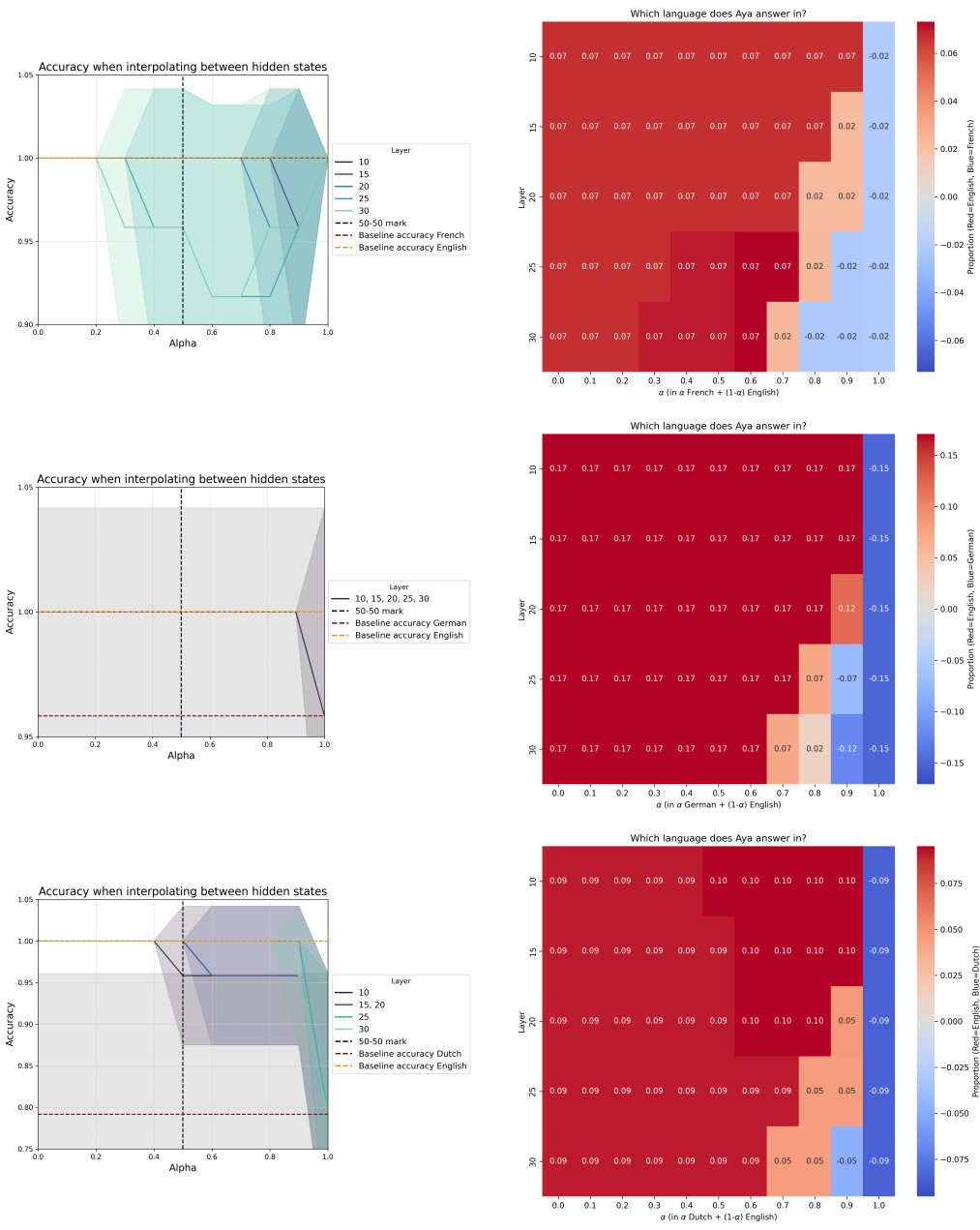

Figure 14: Aya-23-35B Interpolation results in English.

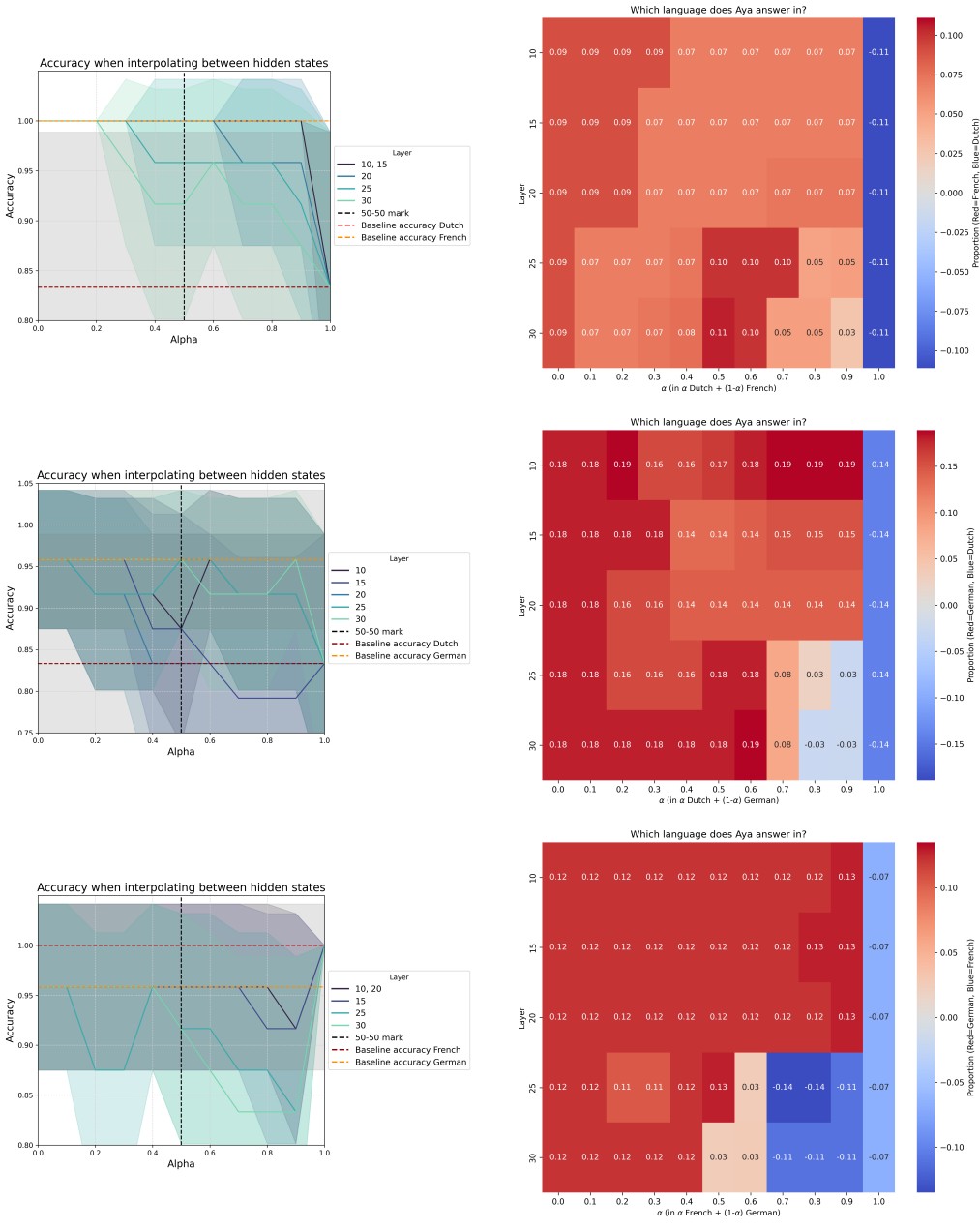

Figure 15: Aya-23-35B Interpolation results in English.

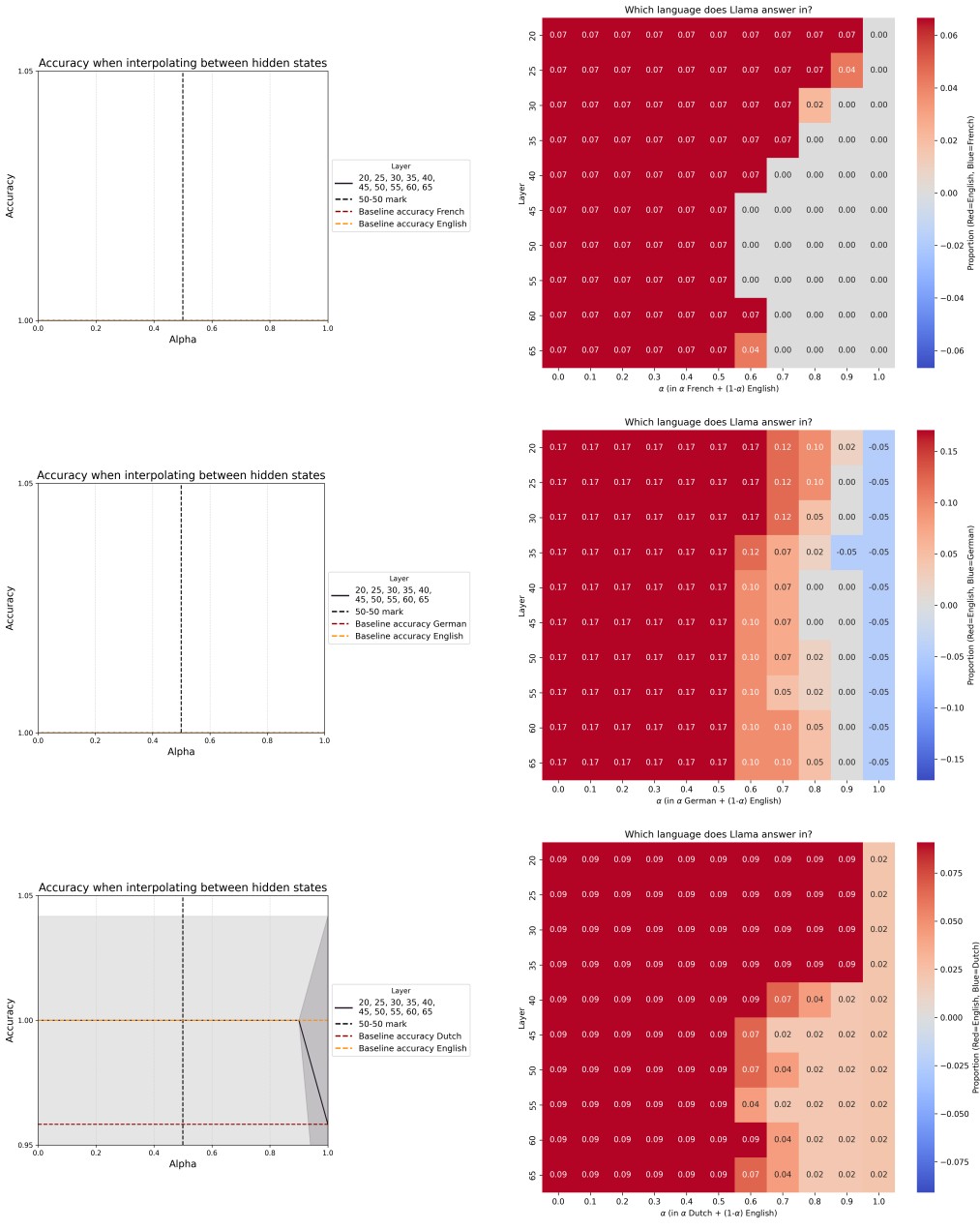

Figure 16: Llama-3.1-70B Interpolation results in English.

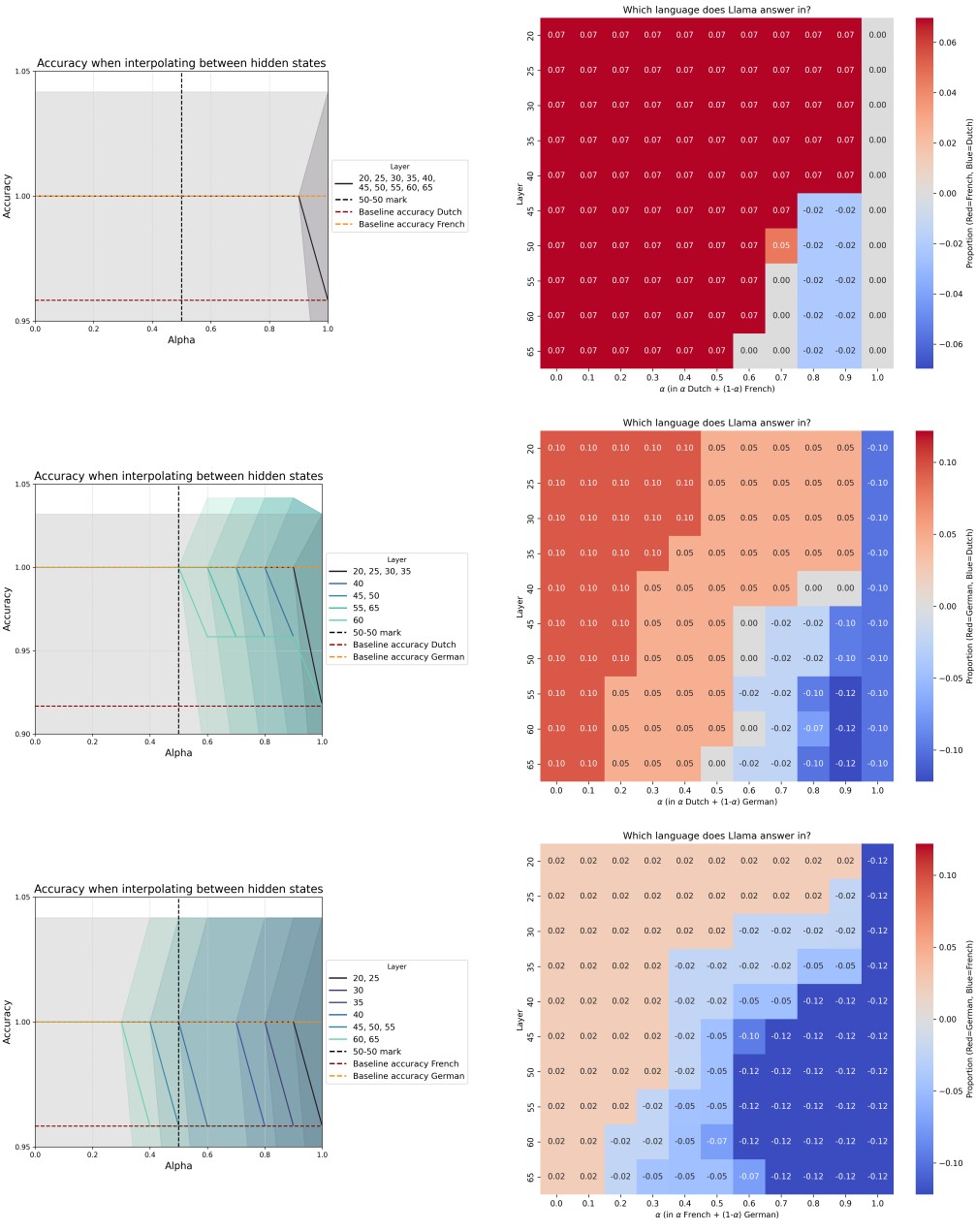

Figure 17: Llama-3.1-70B Interpolation results in English.

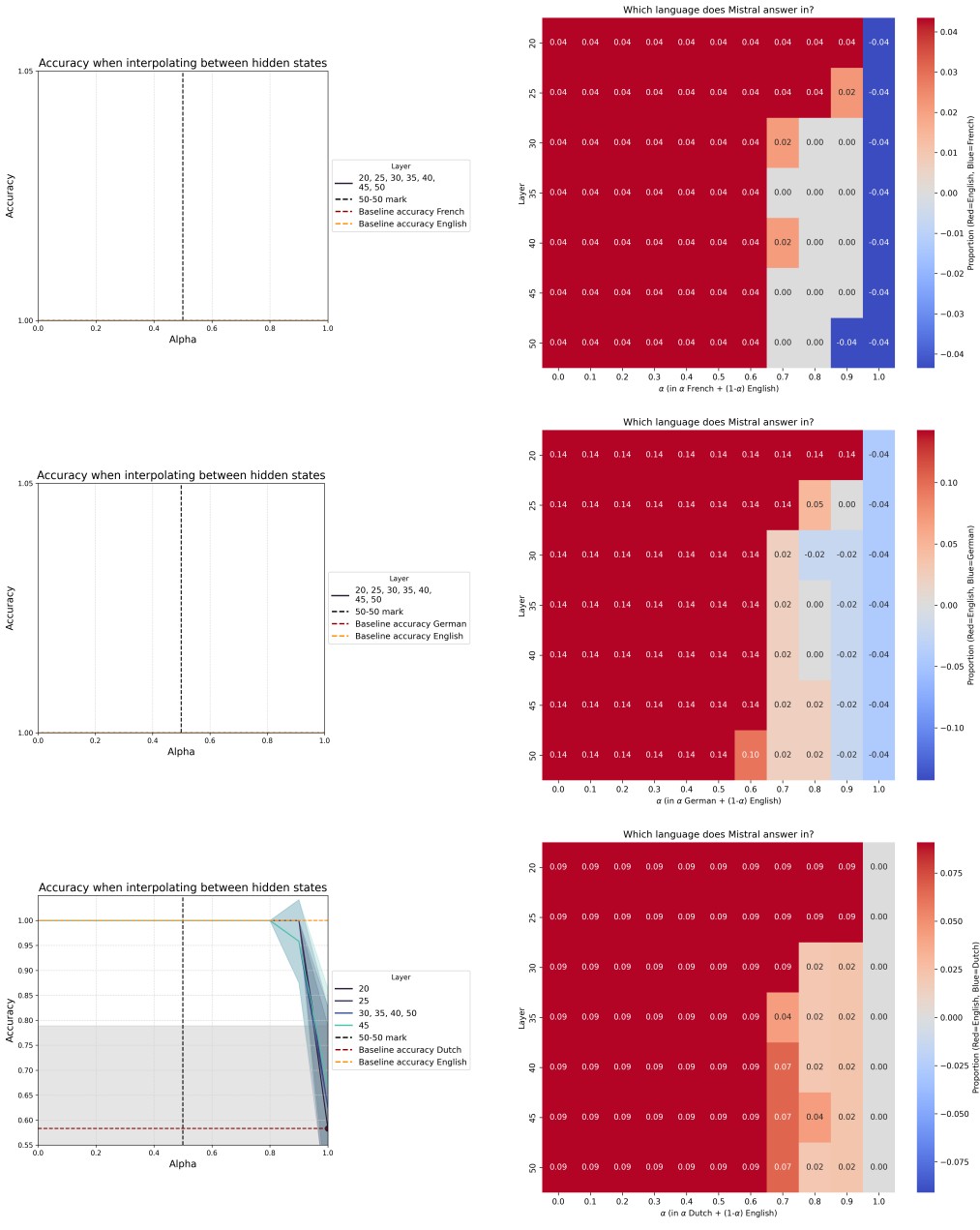

Figure 18: Mixtral-8x22B Interpolation results in English.

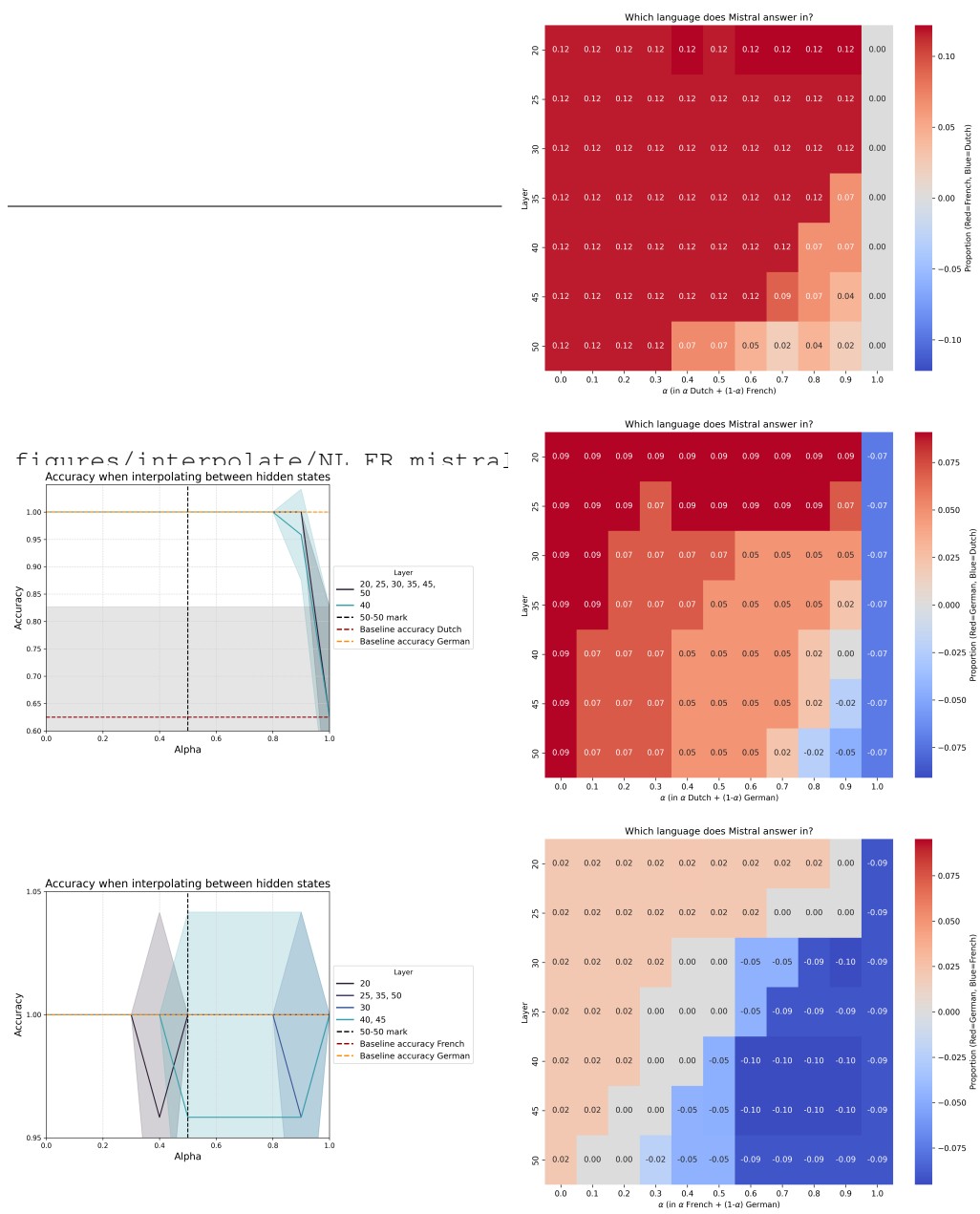

figures/interpolate/NL_FR_mistral

Figure 19: Mixtral-8x22B Interpolation results in English.

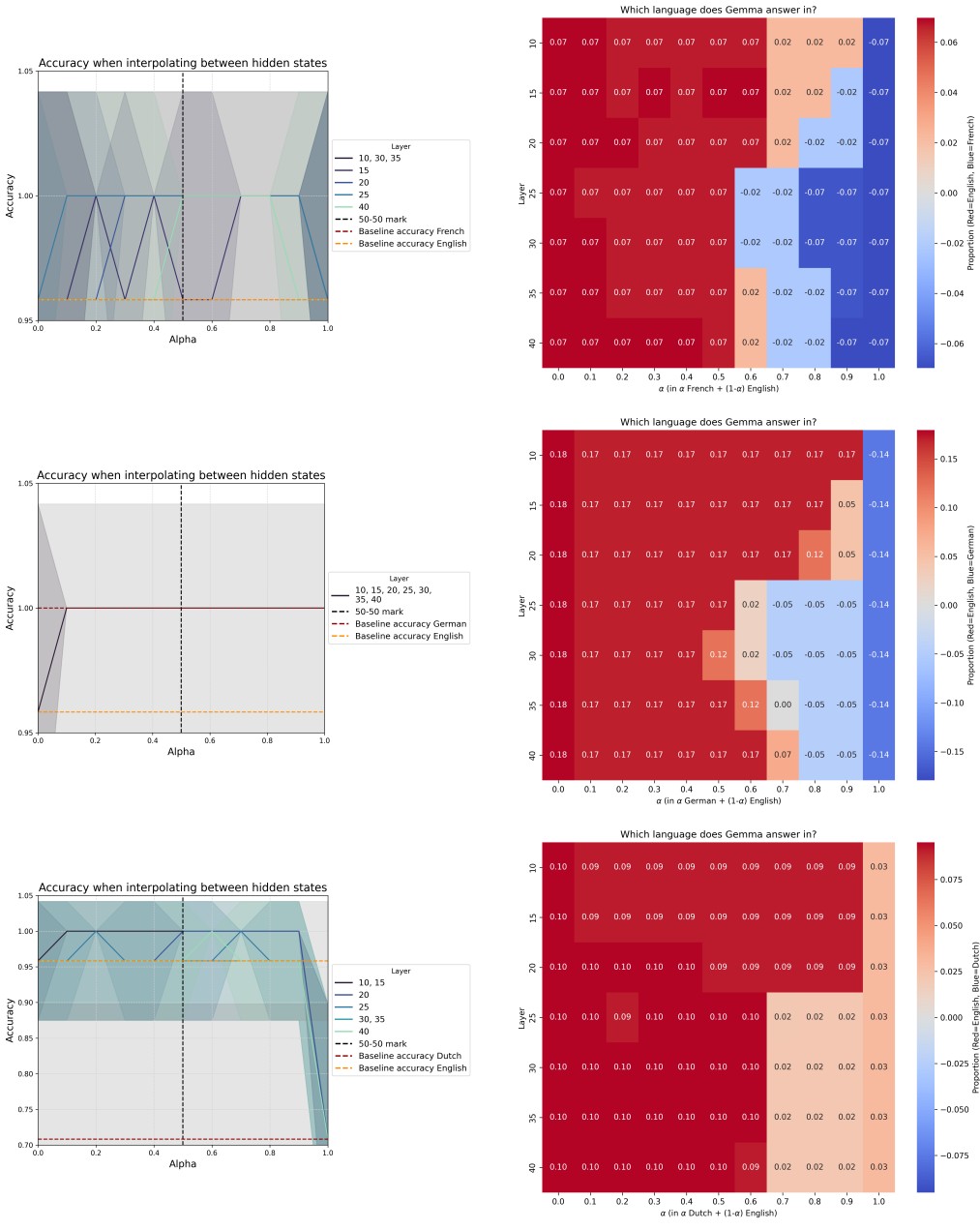

Figure 20: Gemma-2-27b Interpolation results in English.

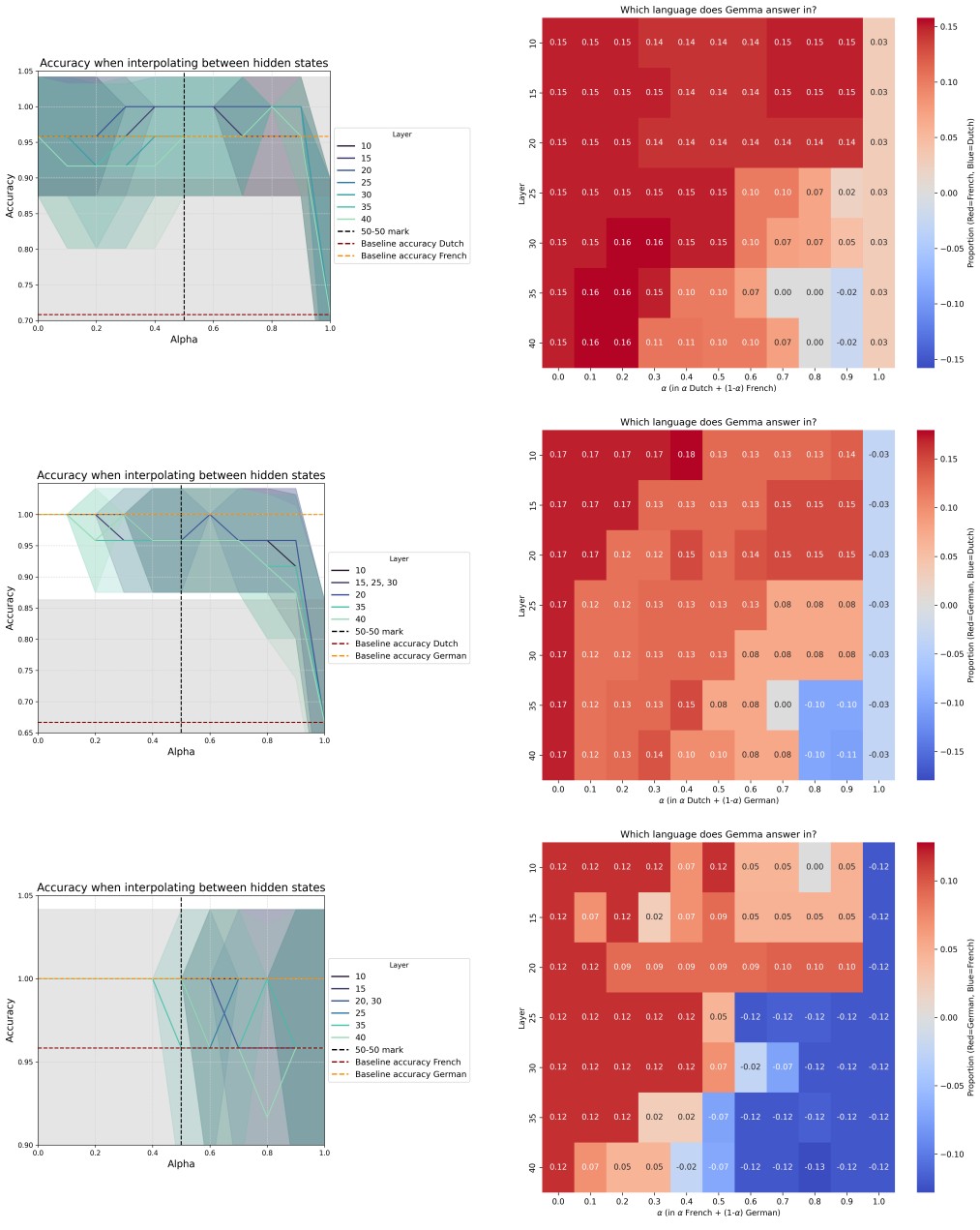

Figure 21: Gemma-2-27b Interpolation results in English.

