# OpenReview forum: "Do Multilingual LLMs Think In English?"
_ICLR.cc/2025/Workshop/BuildingTrust — BuildingTrust_

### Official Review · Reviewer_FhRa · 2025-02-26
**Finds more evidence for an decently well backed hypothesis - that LLMs have English-centered concept spaces - with sound experiments**

**Rating:** 6
**Confidence:** 3

**Review:**

This paper provides further evidence of English centered concept spaces in LLMs predominantly pre-trained on English.

**Quality**

Pros:
- Experiments are generally sound and have valid interpretations, compelling evidence
- Experiments provide consistent results

Cons:
- A few typos, grammatical, and semantic mistakes
- Related works section was only loosely relevant, it was also near the end, which is an unusual choice.  It did not cover existing papers in  the 'in what language do LLMs reason'  line of study, which was confusing.
- Would've been interesting to see whether 'reasoning in language X' is correlated with the amount of pre-training text in that language, though I don't know whether that data is publicly accessible.
- One of the mandarin prompts

**Clarity**

Pros:
- Abstract and introduction are clear and concise, describes paper well
- Figures offer clear illustration of results
- Cites relevant work
- The breakdown between different elements of diction is interesting!

Cons:
- The main body of the paper needs more detail on experimental methodology as they are unclear.  What data was used to generate steering vectors (example pair), which steering setup was used (CAA?).  Briefly, how was the dataset created?  Minor issue.
- Figures need clear descriptions, for example, figure 5 was never referenced in main text

**Originality**

I'm not confident in this evaluation as I'm not super familiar with this line of study, but it seems that similar experiments have been done in various cited work that support the same hypotheses.  The only novel contribution is studying latent representations with multi-token open ended generation, which I don't think contributes *significant* new evidence for the question being studied.

**Significance**

To my knowledge, the hypothesis "LLMs ‘operate’ in a space that is English-centric (or centered on the main pretraining language)" already well backed by evidence.  So in that respect, I'm unsure of this paper's significance, although it does study the problem in a new setting.  The implications of this study are not well formulated, the 'so what?' question is not really answered, arguably because I think this result currently has limited applications...?

---

### Official Review · Reviewer_oGNM · 2025-03-02
**Good paper, accept**

**Rating:** 8
**Confidence:** 3

**Review:**

This work investigates language models from the perspective of pretraining data language composition. Given the dominance of English in current research, it specifically focuses on the English-centric characteristics of LLMs. It uses interp tools to test out three hypotheses about the relation between the languages, providing insights into the behaviors of LLMs and their implications on fairness and performance.

Pros:
- Robust experimental design: Employs a comprehensive approach using logit lens, vector steering and causal tracing to dissect the internal representations of LLMs,
- Model Selection: They analyse four open-weights models (Llama, Gemma, Mixtral, and Aya), which vary in architecture and language coverage. Models like Aya are more multilingual than the Gemma models based on their pretraining data. The paper's results agree with this.
- The methods and dataset are transparent and seem sound, providing a decent foundation for future work. Interesting findings like English steering vectors being more effective definitely open up research directions.
- Multilingual models are the next paradigm of language modeling and this is a meaningful contribution to making them more understandable and powerful

Improvements:
- Limited language coverage: All the languages are high resource, might have been interesting to see how a low-resource language benefits or gets impacted (perhaps giving meaningful results towards testing out whether low-resource language can benefit directly from high-res ones in the same model).
- They mention tokenization as limitation (and this could also explain why they do not include low-resource languages, which obviously have higher fertility and issues with token embedding to word embedding conversions), but do not address it in detail
- Could benefit from more exhaustive analyses and from better contextualization of real-world implications of the results

---

### Official Review · Reviewer_3cF1 · 2025-03-02
**Do Multilingual LLMs Think In English?**

**Rating:** 7
**Confidence:** 4

**Review:**

This paper explores whether multilingual LLMs process information in English, even when given input in other languages. The authors use different interpretability techniques, such as the logit lens, causal tracing, and activation steering, to analyze how LLMs represent language internally. Their findings suggest that models tend to process information in an English-like way before translating it into the output language.

Strengths
Clear Research Question: The study investigates an interesting and important issue in multilingual AI.
Good Methodology: The authors use multiple models, languages, and analysis methods, making the results more reliable.
Strong Results: The findings are well-supported by both quantitative and qualitative evidence.
Important Implications: This study highlights biases in multilingual LLMs, which could affect fairness in AI.

Weaknesses
Limited Language Scope: The paper mainly tests European and Mandarin languages. It would be useful to see results for a more diverse set of languages.
Lack of Dataset Analysis: It does not explore whether this English bias comes from how the models are trained.
No Clear Solutions: The paper does not suggest ways to reduce this bias, which would be helpful for improving multilingual models.

---

### Decision · Program_Chairs · 2025-03-04

**Decision:**

Accept

**Comment:**

The reviewers generally find the paper well-executed and methodologically strong, with clear findings and important implications for multilingual AI fairness. However, concerns remain regarding limited language coverage (focusing on high-resource languages), lack of dataset analysis to explain the bias, and insufficient discussion of real-world impact. Additionally, some prior work on multilingual reasoning is not well covered, and the novelty of findings compared to existing literature is questioned.